# Fermented Lettuce Extract Induces Immune Responses through Polarization of Macrophages into the Pro-Inflammatory M1-Subtype

**DOI:** 10.3390/nu15122750

**Published:** 2023-06-14

**Authors:** Bo-Young Kim, Ji Hyeon Ryu, Jisu Park, Byeongjun Ji, Hyun Soo Chun, Min Sun Kim, Yong-Il Shin

**Affiliations:** 1Research Institute for Convergence of Biomedical Science and Technology, Pusan National University Yangsan Hospital, Yangsan 50612, Republic of Korea; kimboyoung@pusan.ac.kr (B.-Y.K.); wlgus9217@naver.com (J.H.R.); jiss5022@naver.com (J.P.); 2HumanEnos LLC, Wanju 55347, Republic of Korea; wandukong5@naver.com (B.J.); 119bio@naver.com (H.S.C.); 3Center for Nitric Oxide Metabolite, Wonkwang University, Iksan 54538, Republic of Korea; mskim@wku.ac.kr; 4Department of Rehabilitation Medicine, School of Medicine, Pusan National University, Yangsan 50612, Republic of Korea

**Keywords:** lettuce, M1/M2 macrophages, tumor-associated macrophages

## Abstract

It has been reported that lettuce and its bioactive compounds enhance the host immune system by acting as immune modulators. This study aimed to identify the immunological effect of fermented lettuce extract (FLE) on macrophages. To evaluate the efficacy of FLE in enhancing macrophage function, we measured and compared the levels of macrophage activation-related markers in FLE- and lipopolysaccharide (LPS)-stimulated RAW 264.7 cells. Treatment with FLE activated RAW 264.7 macrophages, increased their phagocytic ability, and increased the production of nitric oxide (NO) and pro-inflammatory cytokine levels—similar to LPS. The effects of FLE on M1/M2 macrophage polarization were investigated by determining M1 and M2 macrophage transcript markers in mouse peritoneal macrophages. The FLE-related treatment of peritoneal macrophages enhanced the expression of M1 markers but reduced IL-4 treatment-induced M2 markers. After the generation of tumor-associated macrophages (TAMs), alterations in the levels of M1 and M2 macrophage markers were measured after treatment with FLE. The FLE-related treatment of TAMs increased the expression and production of pro-inflammatory cytokines and also led to the enhanced apoptosis of pancreatic cancer cells. These findings suggest that FLE may be useful for macrophage-targeted cancer therapy because of its ability to regulate the activation and polarization of macrophages in the tumor microenvironment.

## 1. Introduction

Lettuce (*Lactuca Sativa* L.) is one of the most commonly consumed leafy vegetables in the world. Lettuce contains natural phytochemicals, which are essential nutritional bioactive compounds, such as polyphenols, carotenoids, and chlorophyll [1]. It has been reported that bioactive compounds extracted from lettuce have diverse applications [2]. The unknown compounds in lettuce extracts also exhibit various biological properties; several studies have reported the effects of water-soluble extracts from green lettuce on macrophage activation [3], the antidiabetic effects of extracts from fermented lettuce [4], the and palliative effacement effects of fermented soybean-lettuce powder against menopausal symptoms [5]. A previous study of ours reported that fermented lettuce extract (FLE) can improve clinical symptoms of rheumatoid arthritis in mouse models and inhibit the migration and proliferation of the human fibroblast synoviocyte in vitro [6]. These findings suggest that extracts or ferments from lettuce have bioactive compounds that could ameliorate diverse diseases; however, little is known the mechanism of FLE and its effects on various macrophage subtypes.

Macrophages are responsible for innate immunity and induce inflammatory responses by presenting antigens to lymphocytes. They exhibit various phenotypes and can be predominantly divided into the following two major categories: classically activated M1 and alternatively activated M2. Macrophages can be polarized into M1 through treatment with interferon γ (IFN-γ), tumor necrosis factor α (TNF-α), or lipopolysaccharide (LPS), and they play key roles in innate host defense and kill tumor cells by secreting pro-inflammatory cytokines [7]. In contrast, M2 macrophages are activated by exposure to Th2 cytokines, such as interleukin (IL)-4, IL-10, or IL-13 [8,9]. These are associated with humoral immunity, wound healing, and tissue remodeling and also contribute to tumor development by producing anti-inflammatory cytokines such as IL-10, IL-13, and transforming growth factor (TGF)-β [9]. A range of macrophage subtypes play various critical roles in homeostatic and immune responses; however, they have not been fully characterized [10]. Macrophages play important roles in cancer and normal physiological conditions [10]. In most established tumors, tumor-associated macrophages (TAMs) are considered M2-skewed macrophages exhibiting typical features of M2 subtypes, and these possess exert pro-tumor effects, such as promoting tumorigenesis, immunosuppression, and accelerating metastasis. Therefore, targeting the reprogramming of TAMs toward anti-tumor M1-like macrophages is an efficient anti-cancer strategy [11]. These findings indicate that an effective anti-cancer therapeutic strategy involves identifying materials that could activate macrophages and reprogram TAMs.

Pancreatic ductal adenocarcinoma (PDAC) accounts for over 85% of all pancreatic cancer cases and is a highly lethal malignancy with an average 5-year survival rate of less than 10% worldwide [12]. TAMs are one of the most abundant immune cell populations in the pancreatic tumor stroma and are closely involved in PDAC progression [13]. Therefore, to improve anti-tumor immunity and inhibit PDAC progression, it is vital to inhibit the production of M2 macrophages in the tumor stroma [13].

This study aimed to investigate whether FLE influences the effects of macrophages at the molecular and cellular levels. Here, we report a novel pharmacological action of FLE that involves the activation and polarization of macrophages, as indicated by the upregulation of the expression of inflammatory molecules as well as increased levels of cell surface and functional modifications. Additionally, we determined the effects of FLE on the reprogramming of TAMs and the survival of pancreatic cancer cells.

## 2. Materials and Methods

### 2.1. Preparation of FLE

FLE was manufactured by Human Enos (Wanju-gun, Republic of Korea). Briefly, lettuce was mixed with distilled water in a 1:1 ratio; then, it was fermented at 30 °C for 21 days in optimized conditions, including aeration, temperature, and pH. At the end of 21 days, the liquid product was obtained through separation with a centrifuge [5].

### 2.2. Cells Culture and Isolation of Mouse Peritoneal Macrophages

RAW 264.7, MIA Paca-2, and THP-1 cells were purchased from the American Type Culture Collection (ATCC; Manassas, VA, USA). RAW 264.7 and THP-1 cells were maintained in Roswell Park Memorial Institute 1640 medium (RPMI 1640; Welgene, Gyeongsan, Republic of Korea) and supplemented with 10% fetal bovine serum (FBS; Welgene) and 1% penicillin/streptomycin (Welgene). MIA Paca-2 cells were maintained in Dulbecco’s modified Eagle’s medium (DMEM; Welgene) and supplemented with 10% FBS, 5% horse serum (Gibco, Carlsbad, CA, USA) and 1% penicillin/streptomycin (Welgene). The cultures were incubated in a humidified atmosphere containing 5% CO_2_ at 37 °C.

To isolate mouse peritoneal macrophages, mice (C57BL6, male, 6 weeks old; Koatech Co., Pyeongtaek, Republic of Korea) were sacrificed 3 days after intraperitoneal injection of 5 mL of 3% (*w*/*v*) brewer thioglycolate medium (Millipore, Burlington, MA, USA). To harvest the peritoneal cells, ice-cold phosphate-buffered saline (PBS) was injected into the mouse abdomen and then the fluid was withdrawn. After centrifugation, the cells were counted and plated at a density of 1 × 10^6^ cells/mL in RPMI 1640 medium containing 10% FBS and 1% penicillin–streptomycin. To isolate the macrophages, non-adherent cells were gently washed three times with warm medium 18 h after plating [14]. To differentiate macrophages, the cells were treated with LPS (1 μg/mL; Sigma-Aldrich, St. Louis, MO, USA), mouse recombinant IL-4 (10 ng/mL; Peprotech, Cranbury, NJ, USA), or FLE (2%) for 24 and 48 h [15].

To generate TAMs, THP-1 cells (5 × 10^5^ cells/mL) were stimulated with phorbol-12-myristate-13-acetate (PMA; 150 nM; Sigma-Aldrich) for 24 h, removed PMA by washing, and then replaced with fresh media daily. After 2 days, the cells were cultured in the supernatant mix and conditioned in media from MIA Paca-2 cells and fresh complete media for 72 h [16]. The cells were incubated in the presence/absence of FLE, and further analysis was conducted. TAMs and pancreatic cancer cell lines were co-cultivated using a non-contact co-culture transwell system (SPL Life Sciences, Pocheon, Republic of Korea). Inserts containing TAMs were transferred to 6-well plates seeded with MIA Paca-2 cells (4 × 10^5^ cells/well) and co-cultured [17]. After 48 h of co-culture, MIA Paca-2 cells were harvested for further analysis.

### 2.3. Cell Viability

Cell viability was analyzed as previously described [6]. After seeding the cells in 96-well plates, they were treated with FLE and incubated for 24 h after 3-(4,5-dimethylthiazol-2-yl)-2,5-diphenyltetrazolium bromide (MTT; 5 mg/mL; Duchefa, Haarlem, Netherlands) was added to each well. They were incubated for 2 h; then, the supernatant was aspirated and 100 μL of DMSO was added to dissolve the remaining formazan crystals in each well. The optical density per well was measured at 570 nm using a microplate reader (Tecan, Infinite M200, Austria).

### 2.4. Enzyme-Linked Immunosorbent Assay (ELISA)

The levels of mouse IL-1β/12, TNF-α, and IL-6 and human IL-1β and TNF-α secreted in the culture media were determined using commercially available ELISA kits according to the manufacturer’s instructions (R&D Systems, Minneapolis, MN, USA). Briefly, standards and samples (cultured medium) were added to each well of the capture antibody-precoated plates and incubated. After washing the plates, those were incubated with horseradish peroxidase (HRP)-conjugated antibodies. The plates were washed again and incubated with substrate solution under light-protected conditions to induce colorimetric reaction. The reaction was stopped by stop solution, and then absorbance was measured at 450 nm using a microplate reader (Tecan).

### 2.5. Phagocytosis Assay

The phagocytosis assay was performed as previously described [18]. RAW 264.7 cells were stimulated with LPS or FLE in an imaging dish. The latex bead-rabbit IgG-FITC complex (Cayman Chemical, Ann Arbor, MI, USA) was added at a dilution of 1:10 in the culture medium and incubated for 2 h. After washing the cells with assay buffer, the degree of phagocytosis was analyzed using a confocal laser-scanning microscope (K1-Fluo; Nanoscope Systems, Daejeon, Republic of Korea), equipped with a fluorescence system.

### 2.6. Reverse Transcription (RT)-Polymerase Chain Reaction (PCR) and Real-Time PCR

Total RNA was purified from cells using the TRIzol reagent (Invitrogen, Waltham, MA, USA); then, reverse transcription of total RNA was conducted at 42 °C for 1 h with the antiRivert Platinum cDNA synthesis master mix (GenDepot, Barker, TX, USA) and real-time PCR was performed in triplicate using a LightCycler 96 Real-time PCR System (Roche, Basel, Switzerland) [19]. The thermal cycling conditions were as follows: 95 °C for 15 min, 40 cycles of denaturation at 95 °C for 20 s, annealing at 55 °C for 40 s, and elongation at 72 °C for 30 s. The primers used are listed in Table 1. The relative expression of each gene was calculated as the ratio to the housekeeping gene using LightCycler^®^ 96 software (version 1.1.0.1320; Roche). The levels of target gene mRNA were normalized to those of *GAPDH* using the 2^−ΔΔCt^ method [20].

### 2.7. Western Blotting

Western blotting was performed as previously described [6]. The cells were lysed with IP lysis buffer (Thermo Fisher Scientific, Rockford, IL, USA) containing protease and phosphatase inhibitor cocktails (GenDEPOT) following manufacturer protocol. The sample was centrifuged for 15 min at 16,609× *g*, and 4 °C. Protein concentrations were determined using Bicinchoninic Acid Protein Assay Kit (Thermo Scientific). The samples were loaded at equal concentration and separated using 8–15% sodium dodecyl sulphate polyacrylamide gel electrophoresis (SDS-PAGE) and transferred to polyvinylidene difluoride membranes (PVDF, Millipore, Darmstadt, Germany). After blocking for 1 h in 5% skim milk (in TBS containing 0.1% Tween-20; TBS-T), membranes were incubated with primary antibodies against phosphorylated NF-κB p65 (1:1000; Cell Signaling, Danvers, MA, USA), NF-κB p65 (1:1000; Cell Signaling), PARP (1:1000; Cell Signaling), caspase-3 (1:1000; Cell Signaling), mutant p53 (1:1000; Abcam, Cambridge, MA, USA), or β-actin (1:5000; Sigma-Aldrich) at 4 °C overnight. After three washes with TBS-T, the membranes were incubated for 1 h with the corresponding HRP-conjugated secondary antibodies (1:50,000; ENZO Life Sciences, Farmingdale, NY, USA) at room temperature. The bands were reacted using chemiluminescent detection reagents (Amersham Pharmacia, Piscataway, NJ, USA), and developed chemiluminescent intensity was detected by Amersham^TM^ ImageQuant^TM^ 800 (Cytiva, Marlborough, MA, USA).

### 2.8. Apoptosis Assay

An apoptosis assay was conducted using the Annexin V-FITC apoptosis detection kit according to the manufacturer’s instructions (BD Biosciences, Franklin Lakes, NJ, USA). After 48 h of being co-cultured with TAMs in the presence/absence of FLE, MIA Paca-2 cells were harvested. The cells were washed with PBS and resuspended in a binding buffer. Annexin V-FITC and propidium iodide (PI) were added, respectively, and incubated for 15 min at room temperature in the dark. After washing the cells with binding buffer, they were resuspended in binding buffer and analyzed by using FACS Canto II flow cytometry (BD Biosciences).

### 2.9. Statistical Analysis

One-way ANOVA followed by Dunnett’s multiple comparison test were performed using PRISM (version 5.0) (GraphPad Software Inc., San Diego, CA, USA). Results were considered statistically significant at *p*-values < 0.05.

## 3. Results

### 3.1. Effects of FLE on the Activation of RAW 264.7 Macrophages

To determine the cytotoxic effects of FLE on RAW 264.7 cells, the cells were exposed to FLE (0–20%) for 24 h. The viability assay showed that FLE had no significant cytotoxic effects at concentrations up to 2% compared with the control (Figure 1A). A significant dose-dependent decrease in cell viability was observed after exposure to FLE at concentrations of 4–20% (Figure 1A). Accordingly, we used concentrations of FLE <2% in subsequent experiments.

Therefore, we evaluated the effects of FLE on macrophage activation. The morphology of RAW 264.7 cells was altered from the original round shape to a rough form with pseudopodia—a macrophage activation signal—after treatment with FLE, similar to that observed in LPS-stimulated cells (Figure 1B). FLE treatment increased nitric oxide (NO) production in a dose-dependent manner (Figure 1C). These effects were stronger than those exerted by LPS, which is used to activate macrophages. The stimulation of macrophages with FLE resulted in the increased expression of molecules associated with the M1 phenotype, such as TNF-α, IL-1, and IL-6, and which is comparable to that obtained with LPS (Figure 1C). We further assessed whether the phagocytic function of RAW 264.7 macrophages resulted in an uptake of latex bead-FITC after LPS or FLE treatment. FLE-treated cells were observed with the endosomal loading of latex bead-FITC, similar to LPS-stimulated cells (Figure 1D). These results indicate that FLE stimulates the activation of macrophages, similar to LPS, which alters cell morphology and increases the expression of NO and pro-inflammatory cytokines.

### 3.2. Induction of Polarization into M1 Macrophages upon FLE Treatment

To investigate the effects of FLE on M1/M2 macrophage polarization, the expression levels of M1 and M2 macrophage transcript markers were measured in mouse peritoneal macrophages treated with LPS, IL-4, or FLE. Isolated peritoneal macrophages were verified through staining with CD11b, a macrophage marker, and CD11b-positive cells accounted for >95.0% of the cells. Mouse peritoneal macrophages treated with LPS for M1 polarization exhibited a morphology with large filopodia, whereas IL-4-treated macrophages retained a spindle or round shape (Figure 2A). Furthermore, macrophages exposed to FLE showed an increased expression of M1 markers (*Nos2*, *Tnf-α* and *Il-1β*) associated with pro-inflammatory effects. However, the expression of M2 markers (*Arg-1*, *Cd206*, and *Il-10*) was downregulated by FLE (Figure 2B).

We further determined the effects of FLE on IL-4-induced M2 macrophages through evaluation of M2 marker mRNA levels. IL-4-induced M2 macrophages remained spindle- or round-shaped, unlike the dendritic-like shaped FLE-treated macrophages, and the number of these cells gradually increased in a time-dependent manner (Figure 2C). Furthermore, M2 macrophages exposed to FLE showed a reduced expression of M2 markers (Figure 2D). These results suggest that exposure to FLE could lead to the polarization of macrophages into M1-subtypes.

### 3.3. Effects of FLE-Reprogrammed TAMs on Pancreatic Cancer Cells

We investigated whether FLE altered the characteristics of TAMs to those of M1 macrophages. After the generation of TAMs, these cells were exposed to FLE, and the presence of specific markers of M1 macrophages were confirmed. TAMs exposure to FLE enhanced the expression of M1-specific markers, IL-1β, and TNF-α at the mRNA and secreted protein levels (Figure 3A,B). Since NF-κB (a key transcription factor involved in M1 polarization) is required for induction of a number of pro-inflammatory cytokines [21], NF-κB expression in TAMs was compared to that in FLE-treated TAMs. Phospho-NF-κB was not detected in TAMs; however, FLE-exposed TAMs exhibited gradually increasing levels of phosphorylation of NF-κB in a time-dependent manner (Figure 3C). These results indicated that FLE could reprogram TAMs into M1-like phenotypes via NF-κB signaling.

To confirm the effects of FLE-exposed TAMs on pancreatic cancer cells, MIA Paca-2 cells were co-cultured with TAMs in the presence or absence of FLE. Western blotting revealed that there was no significant change in the expression of apoptosis signaling molecules in MIA Paca-2 cells co-cultured with TAMs compared to cells cultured alone; however, FLE-reprogrammed TAMs induced the apoptosis signaling pathway (Figure 4A). The expressions of cleaved PARP and cleaved caspase-3 were upregulated, and the expression of mutant p53 was downregulated in cells co-cultured with TAMs exposed to FLE compared to cells cultured alone or co-cultured with TAMs. Apoptotic (annexin V^+^) cells were also significantly increased in MIA Paca-2 cells co-cultured with TAMs and exposed to FLE (Figure 4B). These results suggest that FLE could induce the reprogramming of TAMs toward M1-like macrophages, causing apoptosis in pancreatic cancer cells.

## 4. Discussion

Macrophage activation is triggered by exposure to bacterial components such as LPS and pro-inflammatory cytokines. Recent studies have reported that plant extracts can activate macrophages [22,23,24]. Activated macrophages trigger immune responses driven by type 1 T helper cells and innate lymphoid cells through the production of pro-inflammatory cytokines and NO [25]. The present study showed that RAW 264.7 cells were activated by FLE, similar to LPS, and exhibited production of NO and pro-inflammatory cytokines (IL-1β, IL-6, and TNF-α) as well as altered morphology (Figure 1B,C). Phagocytosis by macrophages provides the first line of host defense against pathogens and serves as a bridge between innate immune response and the initiation of adaptive immune responses [26]. Therefore, the ability to phagocytize is an important part of the innate immune system as it is critical for the homeostasis of the host [27]. The phagocytic ability of FLE-treated macrophages was also increased (Figure 1D). Macrophages exhibit high plasticity; therefore, they may transition between antagonistic conformations in response to peripheral cues [28]. Using in vitro models of macrophage polarization, we confirmed that macrophages respond to FLE. Similarly, in mouse peritoneal macrophages, treatment with FLE increased the expression of M1 markers, whereas it attenuated the increase in M2 marker gene expression induced by IL-4 (Figure 2B,D). It has been found that morphological differences in macrophage populations could be attributed to their heterogeneity [29]. The addition of FLE in macrophages caused a change to a rough form with pseudopodia similar to the morphology of M1 macrophages (Figure 2A,C). These results suggest that FLE regulates macrophage polarization toward the M1 phenotype.

As mentioned above, M1 polarization is associated with the host defense against several pathogens, tissue injury, and tumor cell death, whereas M2 polarization promotes parasite containment, tissue repair and remodeling, and tumor progression [13,25]. Macrophage duality (a key modulator of local and systemic inflammatory responses) has recently emerged as a therapeutic target for cancer therapy [28]. In the tumor microenvironment, TAMs are disposed to M2 deviation, which promotes and supports tumor behavior by secreting signaling molecules such as growth factors, cytokines, and chemokines [9,30]. Macrophage-targeting strategies for cancer treatment include reprogramming TAMs to activate pro-inflammatory macrophages. The results of this study suggest that FLE can reprogram TAM toward the M1 subtype. In TAMs, phosphorylation of NF-κB—a key transcription factor of M1 polarization—was not induced; however, it was time-dependently upregulated upon FLE treatment (Figure 3C). NF-κB signaling causes the induction of inflammatory genes, including *Tnf-α*, *Il-1β*, *Il-6*, *Il-12p40*, and *Cyclooxygenase-2* [31]. We also confirmed the markedly increased expression and production of IL-1β and TNF-α in TAMs exposed to FLE (Figure 3A,B). FLE may induce the reprogramming of TAMs to M1 macrophages and produce pro-inflammatory cytokines via NF-κB signaling.

NO is an important immune modulator of the tumor microenvironment (TME) and the main chemical that macrophages use to exert an immune attack against tumor antigens [32,33]. It has been reported that treatment with an NO donor (S-nitrosoglutathione) decreased M2-like TAMs and increased M1-like TAMs in a castration-resistant prostate cancer mouse model [34]. This finding indicated that an NO donor could affect the activity of tumor-associated macrophages. Our previous study addressed the fact that FLE consists of numerous active components produced by fermentation as well as NO [5,6]. Although the influence of these components cannot be excluded, we anticipated that NO—components of FLE—would act as an immune modulator on macrophages.

In this study, through the use of a non-contact co-culture system, we demonstrated that the number of apoptotic cancer cells increased after co-culturing with FLE-exposed TAMs. FLE treatment enhanced the production of cytotoxic molecules by TAMs. M1 macrophages can kill target cells directly via mechanisms dependent on ROS/RNS and IL-1β and TNF-α production [35]. In addition to its key role in inflammation, TNF-α can induce cancer cell apoptosis through the tumor necrosis factor receptor 1 (TNFR1) signaling pathway [35]. Once TNF-α binds to TNFR1 receptor, it can induce activation of upstream initiator caspases, which in turn leads to the activation of effector caspases [36]. The expression of IL-1β and TNF-α was upregulated in FLE-treated TAMs (Figure 3B). These molecules may trigger the production of TNFR1, and caspase activation occurs. Activated caspases induce the cleavage of PARP1, which is vital during apoptosis because the degradation of the target protein ultimately leads to biochemical and morphological changes in apoptotic cells [37]. Activated caspase-3 and cleaved PARP were induced in MIA Paca-2 cells co-cultured with FLE-exposed TAMs, and the expression of mutant p53, which can exert anti-apoptotic gain-of-function activity, was downregulated (Figure 4A) [38]. Consequently, the percentage of apoptotic cells increased in MIA Paca-2 cells co-cultured with FLE-exposed TAMs (Figure 4B). These results indicate that reprogramming TAMs via FLE may induce apoptosis in cancer cells by activating macrophages.

We acknowledge the following limitations of our study: First, a flow cytometry analysis of the macrophage polarization status would have further verified the above-mentioned results (Figure 2B,D and Figure 3A). Second, the component analysis of the FLE used in this study was not performed. We have predicted that NO—components of FLE—are substances that activate macrophages. To clarify this limitation, future studies should investigate the properties of FLE components and identify active constituents to macrophages.

In summary, the treatment of macrophages with FLE led to the activation and polarization of M1 macrophages, which produced pro-inflammatory molecules. Moreover, FLE-exposed TAMs exhibited altered characteristics similar to M1 macrophages via NF-κB signaling and induced the apoptosis of pancreatic cancer cells in a non-contact co-culture system (Figure 5). These results can provide a foundation for investigating the action of FLE on macrophages as a novel strategy for pancreatic cancer immunotherapy.

## Figures and Tables

**Figure 1 nutrients-15-02750-f001:**
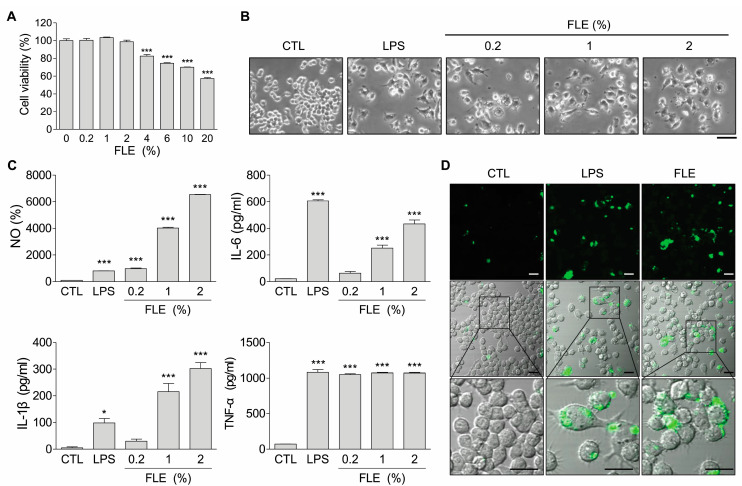
Fermented lettuce extract (FLE) activates macrophages, RAW 264.7 cells. The cells were exposed to lipopolysaccharide (LPS) or FLE (indicated concentration) for 24 h. (**A**) Cell viability was confirmed through the MTT assay. *** *p* < 0.001 compared to CTL (control). (**B**) Changes in cell morphology were observed using an inverted microscope. The scale bar represents 100 μm. (**C**) The levels of nitric oxide (NO) production upon FLE treatment were measured. The expression of pro-inflammatory cytokines was assessed using ELISA. * *p* < 0.05, *** *p* < 0.001 compared to CTL. (**D**) Cells cultured in the imaging dish and treated with LPS or FLE and loaded with latex bead-FITC. The degree of phagocytosis was analyzed using a confocal laser scanning microscope. The green color indicates latex bead-FITC. The scale bar represents 20 μm. Data are representative of three independent experiments.

**Figure 2 nutrients-15-02750-f002:**
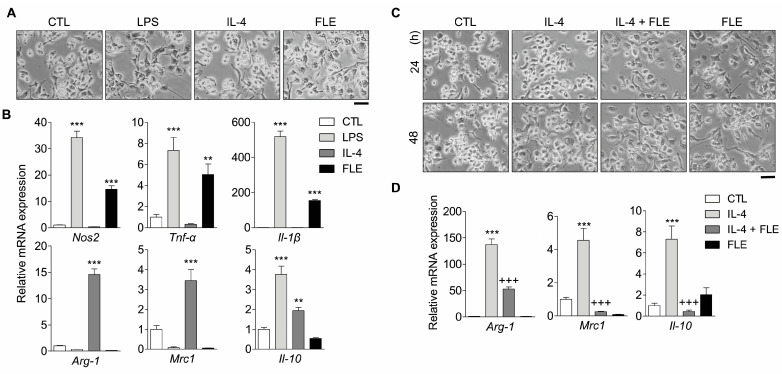
FLE induces the polarization of mouse peritoneal macrophages toward the M1 subtype. The macrophages isolated from mouse peritoneum were treated with LPS (1 μg/mL), IL-4 (10 ng/mL), and/or FLE (2%). (**A**,**C**) Morphology of macrophages visualized using optical microscopy (×200). The scale bar represents 100 μm. (**B**,**D**) Transcripts of M1 (*Nos2*, *Tnf-α* and *Il-1β*) and M2 (*Arg-1*, *Cd206*, and *Il-10*) markers were assessed using real-time PCR. *** *p* < 0.001; ** *p* < 0.01 compared to CTL. +++ *p* < 0.001 compared to IL-4. Data are representative of three independent experiments.

**Figure 3 nutrients-15-02750-f003:**
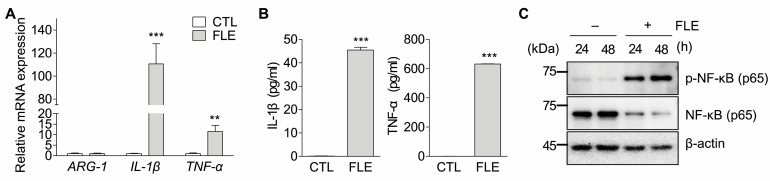
FLE induces reprogramming of M2-like Tumor-associated macrophages (TAMs) into M1 subtype. The TAMs generated PMA-treated THP-1 cells with supernatant mix. After 3 days, the cells were treated with FLE (2%) for 24 and 48 h. (**A**) Total RNAs were isolated from the cells, and transcripts of *ARG-1*, *IL-1β*, and *TNF-α* were amplified by real-time PCR. *** *p* < 0.001; ** *p* < 0.01 compared to CTL. (**B**) Cultured media were isolated, and the levels of secreted protein were measured by ELISA. *** *p* < 0.001 compared to CTL. (**C**) Whole cell extracts were isolated and subjected to immunoblotting for phospho-NF-κB (p65), NF-κB, and β-actin. The minus (−) and plus (+) sign represents treatment without/with FLE, respectively. Data are representative of three independent experiments.

**Figure 4 nutrients-15-02750-f004:**
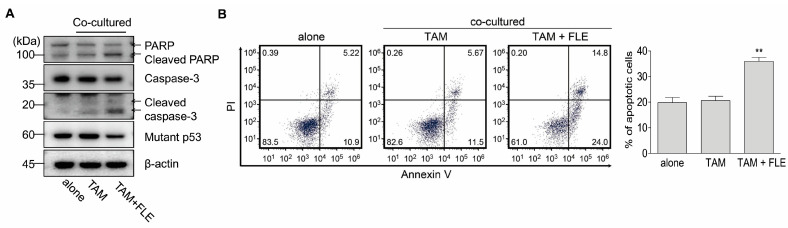
FLE-exposed TAMs cause apoptosis of pancreatic cancer cells. MIA Paca-2 cells were cultured with TAMs in the presence/absence of FLE (2%) for 48 h. (**A**) Cell extracts were obtained after co-culture, followed by western blotting to detect PARP, caspase-3, mutant p53, and β-actin. (**B**) The cells were double-stained with annexin V-FITC and PI and analyzed using flow cytometry. The apoptotic cell percentage data are shown in the graph. ** *p* < 0.01 compared to alone. Data are representative of three independent experiments.

**Figure 5 nutrients-15-02750-f005:**
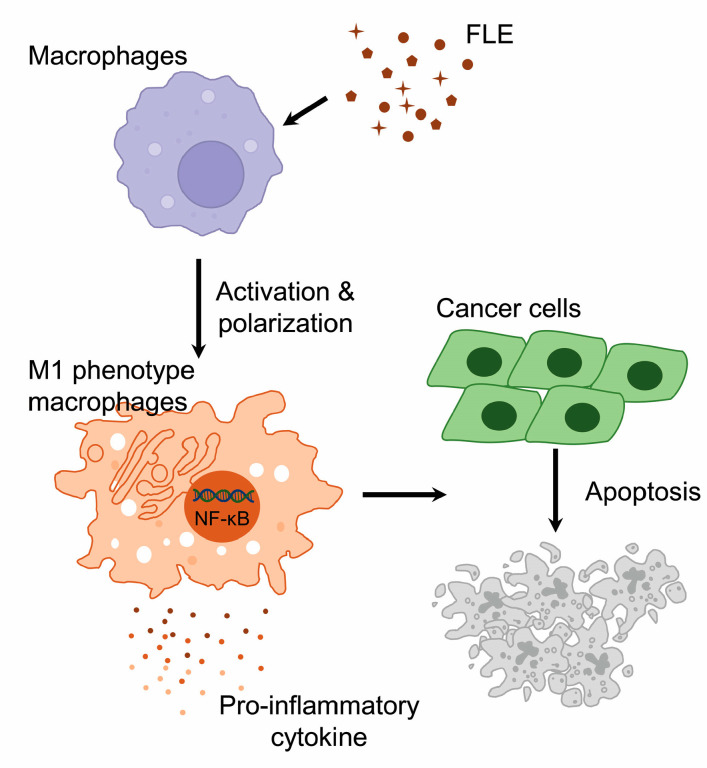
Proposed apoptotic mechanism of cancer cells via FLE-activated macrophages. FLE causes the activation and polarization of macrophages toward M1 phenotype, producing pro-inflammatory molecules via NF-κB signal. The TAMs exposed to FLE induce apoptosis of cancer cells.

**Table 1 nutrients-15-02750-t001:** List of primers used in this study.

Gene	Sequence	
mouse *Gapdh*	Forward	5′-CAGAAGACTGTGGATGGCCC-3′
	Reverse	5′-ATCCACGACGGACACATTGG-3′
mouse *Nos2*	Forward	5′-CGGCAAACATGACTTCAGGC-3′
	Reverse	5′-GCACATCAAAGCGGCCATAG-3′
mouse *Il-1β*	Forward	5′-GCTACCTGTGTCTTTCCCGT-3′
	Reverse	5′-CATCTCGGAGCCTGTAGTGC-3′
mouse *Tnf-α*	Forward	5′-CCTCACACTCACAAACCACCA-3′
	Reverse	5′-GTGAGGAGCACGTAGTCGG-3′
mouse *Arg-1*	Forward	5′-TTTTAGGGTTACGGCCGGTG-3′
	Reverse	5′-TTTGAGAAAGGCGCTCCGAT-3′
mouse *Cd206*	Forward	5′-GGCAAGTATCCACAGCAT-3′
	Reverse	5′-GGTTCCATCACTCCACTC-3′
mouse *Il-10*	Forward	5′-GCTCTTGCACTACCAAAGCC-3′
	Reverse	5′-CTGCTGATCCTCATGCCAGT-3′
Human *GAPDH*	Forward	5′-AAAATCAAGTGGGGCGATGC-3′
	Reverse	5′-GATGACCCTTTTGGCTCCCC-3′
Human *TNF-α*	Forward	5′-CATCCAACCTTCCCAAACGC-3′
	Reverse	5′-CGAAGTGGTGGTCTTGTTGC-3′
Human *IL-1β*	Forward	5′-CGAAGTGGTGGTCTTGTTGC-3′
	Reverse	5′-GGGAACTGGGCAGACTCAAA-3′
Human *ARG-1*	Forward	5′-GTCTGTGGGAAAAGCAAGCG-3′
	Reverse	5′-CACCAGGCTGATTCTTCCGT-3′

## Data Availability

The study did not report any data.

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
