# Peer review of "Fermented Lettuce Extract Induces Immune Responses through Polarization of Macrophages into the Pro-Inflammatory M1-Subtype"

_nutrients, 2023, doi:10.3390/nu15122750_

Round 1

Reviewer 1 Report

The manuscript presented here by Kim et al includes in vitro experiments in which macrophages were stimulated with fermented lettuce extracts or LPS.

The analyses are clear and understandable, but still very simple. One wonders about this simplicity when reading the paper "Fermented Lettuce Extract Containing Nitric Oxide Metabolites Attenuates Inflammatory Parameters in Model Mice and in Human Fibroblast-Like Synoviocytes", published by the same people. 

A few more details on the implied NFkB signaling pathway or phagocytosis would help this paper in terms of content.

Thus, co-cultivations or sequential stimulations of lettuce extracts and LPS would also be conceivable.

The following questions come to mind when I read the manuscript:

Which substances in lettuce induce activation? 

Are cells protected from LPS if they were previously treated with the lettuce extract? 

In general, the figures had very poor resolution, but this may also be due to the generated PDF.

A summary cartoon at the end about the potential mechanism would be beneficial.

Author Response

Thank you for evaluating our manuscript and for your constructive comments and suggestions. We have revised the manuscript accordingly and provided point-by-point responses below.

The manuscript presented here by Kim et al includes in vitro experiments in which macrophages were stimulated with fermented lettuce extracts or LPS.

The analyses are clear and understandable, but still very simple. One wonders about this simplicity when reading the paper "Fermented Lettuce Extract Containing Nitric Oxide Metabolites Attenuates Inflammatory Parameters in Model Mice and in Human Fibroblast-Like Synoviocytes", published by the same people.

Point 1: A few more details on the implied NFkB signaling pathway or phagocytosis would help this paper in terms of content.

Thus, co-cultivations or sequential stimulations of lettuce extracts and LPS would also be conceivable.

Response 1:

  • To investigate whether NF-κB signaling acts as a key transcription factor in TAMs reprogramming by FLE, we confirmed mRNA levels of pro-inflammatory cytokines using an inhibitor for NF-κB (Bay 11-7085). The expression of IL-1β and TNF-α were measured by real-time PCR after treatment with FLE and Bay 11-7085 for 24 h.

The levels of IL-1β and TNF-α mRNA were significantly increased by treatment with FLE, these increases were remarkably decreased by treatment with Bay 11-7085 in TAMs exposed to FLE (see figure below).

  • We conducted a modified treated time; the original treated time with FLE was 48 h, but it has been changed to 24 h. If you give us more time, , we can conduct three independent experiments in the same way as the original experiment method, and then the results can be included in the manuscript.

Point 2: The following questions come to mind when I read the manuscript: Which substances in lettuce induce activation? Are cells protected from LPS if they were previously treated with the lettuce extract?         

Response 2: We sincerely appreciate you pointing this out.

Our previous study addressed that FLE consists of numerous active components produced by fermentation as well as nitric oxide [ref. Park, J.R., et. al., Nutrients 2023, 15, 1106, doi:10.3390/nu15051106.].

For detailed analysis, we tried LC/MS analysis for determining the metabolic characteristics of FLE, but it was difficult to complete it by the scheduled date. We will certainly perform a characteristic analysis of the metabolites of FLE in future studies.

We believe that FLE stimulates the activation of macrophages similarly to LPS. Therefore, FLE does not expect protect cells from LPS in macrophages.

Point 3: In general, the figures had very poor resolution, but this may also be due to the generated PDF.

Response 3: We apologize for the inconvenience. The figures and images were changed to high resolution in the revised manuscript.

Point 4: A summary cartoon at the end about the potential mechanism would be beneficial.

Response 4: We sincerely appreciate the Reviewer’s comments. We added a summary figure (figure 5) to the of the discussion section.

Reviewer 2 Report

Dear Authors,

Thank you very much for your work.

I have some request that should be please addressed.

First, I would like to have each abbreviations written in full + list of abbreviations.

Second, the RT-qPCR should be done with 10 references genes of which the best 3 should be selected. Please read this work: https://pubmed.ncbi.nlm.nih.gov/12184808/

Third, many more details are needed in the methodology.

Fourth, the figures and images need to be larger. For the moment, I caanot read them.

Best wishes,

Reviewer

Dear Authors,

Please check the manuscript for minor spelling errors.

Best wishes,

Reviewer
